# Habitat Suitability Shifts of *Eucommia ulmoides* in Southwest China Under Climate Change Projections

**DOI:** 10.3390/biology14040451

**Published:** 2025-04-21

**Authors:** Qi Liu, Longjiang Liu, Juan Xue, Peiyao Shi, Shanshan Liang

**Affiliations:** 1College of Pharmacy, Guizhou University of Traditional Chinese Medicine, Guiyang 550025, China; liuqik2025@163.com (Q.L.);; 2Provincial Inheritance Base of Traditional Chinese Medicine Processing under National Administration of Traditional Chinese Medicine, Guizhou University of Traditional Chinese Medicine, Guiyang 550025, China

**Keywords:** *Eucommia ulmoides*, climate resilience, species distribution modeling, habitat fragmentation, medicinal plants, karst ecosystems

## Abstract

This study integrates MaxEnt modeling with geospatial analysis to assess future habitat suitability for *Eucommia ulmoides*, a medicinal Tertiary relict species, under SSP1-2.6 and SSP5-8.5 climate scenarios. Key findings reveal precipitation seasonality, thermal extremes, and elevation as primary distribution drivers, with projected southward centroid shifts and habitat contraction in Chongqing. These insights inform conservation strategies for this ecologically vulnerable species.

## 1. Introduction

Climate change serves as a primary driver of plant habitat evolution [1,2,3,4,5]. Projections by the IPCC indicate that under moderate-to-high-emission scenarios (e.g., SSP2-4.5 to SSP5-8.5), global warming will exceed the 1.5 °C threshold during this century [6,7]. This trend is expected to compel endangered plant species to shift toward higher latitudes or altitudes, thereby jeopardizing biodiversity [8]. Analyzing habitat suitability zones is essential for prioritizing biodiversity protection.

Ecological niche models (ENMs) [9] predict species distributions by quantifying habitat suitability or ecological niches. Common ENM methods include MaxEnt [10], GARP [11], and Climex [12]. MaxEnt is widely adopted due to its user-friendly interface, low sample requirements, and high predictive accuracy [13,14]. CMIP6 (Coupled Model Intercomparison Project Phase 6) integrates the Scenario MIP subproject, which includes four Shared Socioeconomic Pathways (SSPs): SSP1-2.6 (low radiative forcing, upgrading RCP2.6 with negative emissions), and SSP5-8.5 (high radiative forcing, exceeding RCP8.5 CO_2_ levels with enhanced non-CO_2_ mitigation efforts) [15,16].

*Eucommia ulmoides* Oliv., a perennial deciduous tree, represents the sole extant Tertiary relict species within the genus Eucommia [17]. Recognized as a rare and endangered species, it holds Class II protected status under China’s conservation legislation [18]. Beyond its renowned pharmacological applications in tonifying liver–kidney functions, enhancing osteogenic capacity, and stabilizing gestational processes, this species serves as a strategic bioresource with multifaceted ecological and economic significance [19]. Its industrial utility spans urban–rural afforestation programs, natural rubber production (notably gutta-percha extraction), and nutraceutical development [20,21].

The native distribution of *Eucommia ulmoides* Oliv. historically centered in Hanzhong (Shaanxi Province) and Yuzhou (Henan Province), with Sichuan, Chongqing, and Guizhou emerging as primary cultivation hubs since the Ming Dynasty (1368–1644) [22]. Since the 20th century, wild populations have faced critical endangerment due to overexploitation driven by traditional medicinal demand, unsustainable silvicultural practices in early plantations, and adverse impacts of pre-industrial resource extraction [23]. In southwestern China, a key native habitat, prolonged deficiencies in systematic conservation and scientific management have accelerated population decline. Natural forest cover reduction, genetic diversity erosion, and ecosystem function degradation further exacerbated this trend.

Previous studies employing the MaxEnt model have simulated the potential habitats and mapped optimal cultivation zones for *Eucommia ulmoides* Oliv. across China [24,25,26]. Historical records trace its native distribution to Hanzhong (modern-day Shaanxi) and Yuzhou (present-day Henan Province), with Sichuan, Chongqing, and Guizhou emerging as dominant cultivation hubs since the Ming Dynasty (1368–1644) [27]. Building on this framework, the present study utilizes the MaxEnt model coupled with ArcGIS technology to forecast potential distributions of *Eucommia ulmoides* Oliv. in southwestern China under SSP5-8.5 and SSP1-2.6 climate scenarios, spanning sequential 20-year intervals (2021–2040, 2041–2060, 2061–2080, and 2081–2100). Through systematic identification of bioclimatic determinants governing habitat suitability, this analysis establishes an evidence-based foundation for optimizing silvicultural practices and adaptive resource governance under evolving climatic regimes.

## 2. Materials and Methods

### 2.1. Introduction to the Research Region

Southwestern China, located at the nation’s southern extremity, extends across geographic coordinates between 97° E and 110° E longitude and 21° N and 34° N latitude. This vast territory comprises four primary administrative divisions: Sichuan Province, Guizhou Province, Yunnan Province, and Chongqing Municipality [28]. The region’s topography features an elevation gradient descending from western to eastern sectors and northern to southern zones, with landscape composition dominated by basin formations and hilly terrains that collectively form one of China’s most geomorphologically diverse regions [29]. Under climate change impacts since the turn of the century, forested areas in this region have demonstrated persistent temperature increases alongside decreasing precipitation patterns and reduced solar radiation exposure [30]. As China’s second-largest contiguous forest ecosystem, this area holds dual recognition as both one of the country’s 25 key biodiversity priority zones and a global conservation hotspot. Ecologically, it maintains crucial functions as a strategic water resource reserve and a national repository for genetic diversity, supporting numerous endemic species [31,32] (Figure 1).

### 2.2. Species Distribution Information

Geospatial occurrence data for *Eucommia ulmoides* were acquired through three national databases: the China National Resources Platform (http://www.nsii.org.cn/2017/home.php, accessed on 13 January 2024), China Digital Plant Herbarium (https://www.cvh.ac.cn/, accessed on 15 January 2024), and China Plant Image Library (https://ppbc.iplant.cn/, accessed on 19 January 2024). Following the elimination of non-compliant and redundant entries [33], 704 validated distribution records were retained, with provincial allocations as follows: Guizhou (119), Sichuan (75), Yunnan (39), and Chongqing (61). Spatial coordinate deduplication enabled generation of the distribution map for southwestern China at 300-unit resolution (Figure 2), with base cartography (1:4,000,000) digitally reproduced from the National Basic Geographic Information System (http://bzdt.ch.mnr.gov.cn/, accessed on 5 December 2024) All georeferenced coordinates were archived in standardized CSV format to facilitate subsequent ecological modeling applications.

### 2.3. Collection and Evaluation of Environmental Data

Climatic and topographic parameters were obtained from WorldClim2.1 (https://worldclim.org/, accessed on 9 July 2024) at 2.5 arcminute resolution, including nineteen bioclimatic variables and elevation data spanning 1970–2000. Slope and aspect layers were calculated using the digital elevation model (Table 1). Future climate projections for four 20-year intervals (2021–2100) were generated through the BCC-CSM2-MR model [34] developed by China’s National Climate Center, employing CMIP6’s SSP1-2.6 and SSP5-8.5 scenarios [35,36] at equivalent spatial resolution. To address potential biases from variable interdependence [24], we implemented a Spearman correlation threshold (r > 0.8) coupled with contribution rate screening, systematically excluding collinear predictors and low-impact factors.

### 2.4. Development of the Model Architecture and Parameter Optimization

The filtered species occurrence records and environmental variables were imported into the MaxEnt 3.4.4 model for analysis. Model configuration included a random test percentage of 25, regularization multiplier of 1, maximum background points of 1000, and maximum iterations of 5000. Model robustness was ensured through 10-fold cross-validation, with environmental variable significance quantified via jackknife tests. All analyses maintained default parameter configurations, generating predictive outputs as ASCII grid files. The ROC curve (receiver operating characteristic curve) is a graphical tool for evaluating classifier performance [37]. It plots the relationship between the true positive rate (TPR) and false positive rate (FPR) across different classification thresholds. Model validation employed receiver operating characteristic (ROC) curve analysis [38], where the Area Under the Curve (AUC) served as the quantitative performance metric. AUC interpretation followed established ecological modeling standards [39]: 0.5–0.6 (marginal credibility); 0.6–0.7 (acceptable certainty); 0.7–0.8 (medium predictive certainty); 0.8–0.9 (high confidence); 0.9–1.0 (exceptional accuracy).

### 2.5. Appropriate Zoning of Eucommia

Habitat suitability modeling outputs from MaxEnt3.4.4 were processed through ArcGIS 10.8 to generate predictive distribution maps across China. Zone classification was based on two criteria: Maximum Training Sensitivity plus Specificity (MTSS) and the Balance between Training Omission, Predicted Area, and Threshold Value (BTPT) [40]. The output raster underwent threshold-based classification using four habitat suitability index (P) categories: non-suitable (*p* < 0.2), low suitability (0.2–0.4), moderate suitability (0.4–0.6), and high suitability (*p* ≥ 0.6).

### 2.6. Projected Centroid Shift of Eucommia ulmoides Under Future Climate Scenarios

Habitat suitability projections from MaxEnt 3.4.4 were processed through ArcGIS 10.8’s Geoprocessing tools to quantify current and future distribution areas for *Eucommia ulmoides*. The SDM framework executed climate scenario analyses using ASCII grid outputs, with habitat viability thresholds calculated via Spatial Analyst functions. Subsequent cartographic visualization integrated classified suitability layers (present/future) through standardized symbology protocols.

## 3. Results

### 3.1. Evaluation and Selection of Environmental Variables

Based on Spearman rank correlation analysis, environmental variables with pairwise correlation coefficients exceeding 0.8 were eliminated to address collinearity. Ten MaxEnt 3.4.4 model runs were performed to identify high-contribution environmental predictors, resulting in eight key factors for final modeling (Table 2). Dominant variables included the following: precipitation of the driest month (Bio14, 27.2%), minimum temperature of the coldest month (Bio6, 22.1%), precipitation of the wettest month (Bio13, 14.9%), annual precipitation (Bio12, 10.4%), mean temperature of the coldest quarter (Bio11, 4.2%), slope (3.0%), mean temperature of the driest quarter (Bio9, 2.2%), and elevation (1.9%). These factors collectively accounted for 85.9% of the model’s explanatory power, significantly exceeding the contributions of other variables.

### 3.2. Evaluation of Model Accuracy

Receiver operating characteristic (ROC) curves for the identical dataset were calculated as mean values from repeated simulations (Figure 3). The mean training AUC across replicates was 0.884 (SD = 0.009), indicating robust predictive performance and high reliability of the *Eucommia ulmoides* potential distribution model.

### 3.3. The Primary Environmental Factors Influencing the Growth of Eucommia ulmoides

Jackknife test results from 10 MaxEnt 3.4.4 model runs identified environmental variable effects (Figure 4). The coldest month’s minimum temperature (Bio6) achieved the highest variable gain when used individually. Conversely, aspect (slope direction) caused the most significant model gain reduction when omitted. Three criteria—contribution rate, permutation importance, and jackknife test results-identified eight key environmental determinants of *Eucommia ulmoides* distribution: precipitation of the driest month (Bio14, 27.2%), minimum temperature of the coldest month (Bio6, 22.1%), precipitation of the wettest month (Bio13, 14.9%), annual precipitation (Bio12, 10.4%), mean temperature of the coldest quarter (Bio11, 4.2%), slope (3.0%), mean temperature of the driest quarter (Bio9, 2.2%), and elevation (1.9%).

### 3.4. Correlation Between Eucommia Distribution and Environmental Factors

Response curves quantify the probabilistic relationship between *Eucommia ulmoides* habitat suitability and environmental factors, with a probability threshold ≥ 0.5 defining optimal habitat ranges (Figure 5). Analysis revealed peak growth suitability under the following conditions: precipitation of the driest month (12.6–96.5 mm), minimum temperature of the coldest month (−4 °C to 6 °C), precipitation of the wettest month (184–345 mm), annual precipitation (216–970 mm), mean temperature of the coldest quarter (0.7–9.8 °C), mean temperature of the driest quarter (3–11.7 °C), and elevation (13.9–756 m) (Table 3). This analysis demonstrates *Eucommia ulmoides* distribution is primarily constrained by precipitation regimes, thermal conditions, and altitudinal gradients.

### 3.5. Projected Distribution Range of Eucommia ulmoides Under Present Climatic Conditions

Empirical field data and geospatial analysis (Figure 2) reveal *Eucommia ulmoides* primarily clusters in mid-elevation valleys (13.9–756 m asl) across Southwest China’s Daba and Wuling mountain systems, with distribution hotspots in central Guizhou and Chongqing municipalities, and marginal occurrences in Yunnan. Ecological niche modeling through MaxEnt3.4.4 integrated with ArcGIS10.8 spatial analysis (Figure 6) identifies the optimal habitats under current bioclimatic conditions as the following: mid-elevation zones (400–1200 m) of the Dabashan karst valleys; moderately elevated plateaus (600–1500 m) in Guizhou’s karst terrain. Moderate suitability areas extend through Sichuan Basin peripheries and eastern Yunnan’s transitional ecotones. These model outputs demonstrate strong congruence with empirical distribution patterns, validating the combined SDM-GIS methodology.

### 3.6. Projected Distribution Range of Eucommia ulmoides Under Future Climate Scenarios

Under the SSP126 climate scenario, *Eucommia ulmoides* exhibited habitat range expansion, while its high-suitability zones contracted toward mountainous areas in northeastern Guizhou under SSP5-8.5 (Figure 7). Projections indicate the following: low-emission scenario (SSP1-2.6): total suitable habitat increases from 28,931 km^2^ (current) to 40,643 km^2^ (2081–2100), with high-suitability areas expanding from 7255 km^2^ to 10,082 km^2^; high-emission scenario (SSP5-8.5): total habitat continues growing, but high-suitability zones decline from 10,633 km^2^ (2061–2080) to 8594 km^2^ (2081–2100), showing irregular 20-year fluctuation patterns (Table 4).

### 3.7. Centroid Shift of Eucommia ulmoides Distribution in High-Suitability Regions Under Future Climate Scenarios

*Eucommia ulmoides* will exhibit minor range centroid shifts under future climate scenarios (Figure 8). Current high-suitability core areas are centered in Chongqing Municipality. Climate projections indicate the following: low-emission scenario (SSP1-2.6): the centroid shifts southwest toward lower latitudes, relocating from Chongqing to Zunyi County; high-emission scenario (SSP5-8.5): the habitat core demonstrates northward movement to higher latitudes.

## 4. Discussion

### 4.1. Response of Potential Suitable Areas for Eucommia ulmoides to Climate Change in Southwest China

Climatic analysis reveals *Eucommia ulmoides* exhibits optimal growth in low-mountain ecosystems (300–500 m asl), particularly in valley slopes with sparse forest cover, under annual precipitation regimes of 400–1500 mm and mean temperatures of 11.0–18.0 °C [41]. However, cultivation remains contraindicated in hydromorphic soils with poor drainage or saline–alkaline substrates [23].

Southwestern China exhibits pronounced climatic heterogeneity, with tropical/subtropical monsoon systems dominating southeastern sectors contrasting northwestern plateau montane regimes [42,43]. Under anthropogenic warming (1961–2008), plateau regions experienced accelerated winter thermal anomalies (+1.2 °C/decade, *p* < 0.05), while basin and hilly landscapes showed moderated warming trends (ΔT < 0.3 °C/decade). Precipitation patterns demonstrate a marked west–east gradient, ranging from 1600 mm annual accumulation in eastern Qinghai–Tibet Plateau to <800 mm in western Sichuan Plateau, with transitional zones in SW/NE Guizhou exceeding 1300 mm [44,45].

### 4.2. Projected Response of the Potential Habitat Area of Eucommia ulmoides in Southwest China to Future Climate Scenarios

Recent decades have witnessed substantial transformations in terrestrial ecosystems, characterized by widespread forest clearance and grassland deterioration, coupled with demographic expansion, urban sprawl, and escalating environmental contaminants, collectively exacerbating greenhouse gas emissions and planetary warming trends [46,47]. Climatic modeling indicates that thermal elevation may amplify atmospheric moisture retention, hasten cryosphere destabilization (encompassing glaciers and permafrost), augment oceanic level rise, and amplify evapotranspiration rates. Furthermore, modifications in hydrologic regimes are predicted to induce temporal–spatial redistribution of precipitation events [48]. Under ongoing climatic warming scenarios, meteorological projections suggest a pronounced escalation of persistent climatic anomalies in southeastern Southwest China from 2025 to 2055, contrasted by a significant reduction in such events across the western Sichuan Plateau [49]. Particularly concerning is the projected rainfall deficit spanning Guizhou, Chongqing, and Yunnan provinces, where diminished precipitation could trigger extended aridity periods with detrimental consequences for phytomass development [50].

### 4.3. The Impact of Climate Change on the Cultivation of Eucommia ulmoides in the Future

As a silviculturally significant species in China, *Eucommia ulmoides* Oliv. exhibits notable ecological plasticity and broad distribution across diverse habitats [51]. Originating from mountainous ecosystems in southwestern China, its core germplasm reserves are concentrated in the southern Qinling Mountain range, encompassing Sichuan, Guizhou, Hubei, and Hunan provinces [52]. Recent decades have witnessed severe depletion of wild populations due to climatic perturbations exacerbated by anthropogenic overexploitation, necessitating a shift toward anthropogenic cultivation. The National Eucommia Industry Development Plan (2016–2030) issued by the National Forestry Administration designates Guizhou, Sichuan, Hunan, and Jiangxi as priority regions for standardized cultivation. This study employs the MaxEnt 3.4.4 ecological niche model integrated with ArcGIS 10.8 spatial analysis to predict optimal habitat suitability and identify critical environmental determinants (e.g., bioclimatic variables, topographic features), thereby establishing scientific guidelines for standardized cultivation protocols and germplasm resource conservation.

The integration of MaxEnt modeling with ArcGIS spatial analysis provides critical insights into habitat suitability dynamics for Eucommia cultivation in Southwest China. Contemporary climatic conditions delineate optimal habitats (probability > 0.6) within mid–low elevation valleys (13.9–756 m ASL) of central Guizhou, Chongqing metropolitan areas, and the Dabashan–Wulingshan mountain ranges. Multivariate analysis identifies precipitation of the driest month (Bio14, 27.2% contribution), minimum temperature of the coldest month (Bio6, 22.1%), and slope gradient (3%) as dominant bioclimatic determinants. Projections under contrasting climate scenarios reveal divergent trajectories: under SSP126 (low radiative forcing), total suitable areas expand to 40,600 km^2^ (2081–2100) with centroid migration toward lower latitudes (e.g., Zunyi County), whereas SSP585 (high emissions) precipitates contraction of high-suitability habitats to 8600 km^2^ accompanied by pronounced northward centroid shifts, underscoring climate-induced spatial heterogeneity in habitat suitability.

These findings inform adaptive silvicultural strategies. Priority conservation zones should target current high-suitability nuclei (e.g., karst landscapes in central Guizhou), implementing optimized cultivation protocols to mitigate drought stress (optimal dry-season precipitation: 12.6–96.5 mm) [53,54,55]. Anticipating centroid migration patterns, climate-resilient cultivation belts spanning southern Chongqing to northeastern Guizhou require establishment, complemented by cold-tolerant cultivar development for marginal zones like the western Sichuan Plateau (cold tolerance threshold: –4 to 6 °C). The altitudinal constraint (<756 m ASL) necessitates the avoidance of high-elevation afforestation while enhancing terrain-specific resource utilization in mid–low elevation hills through microtopographic engineering.

The MaxEnt model exhibits inherent limitations: default parameter selections (e.g., feature classes [FCs], regularization multipliers [RMs]) may elevate model complexity, increasing overfitting risks and reducing prediction accuracy [56]. While optimization via parameter-specific toolkits is feasible, highly complex models paradoxically demonstrate improved species distribution fitting within training zones [57]. Critically, the model lacks universally applicable optimization criteria, necessitating case-specific parameter calibration.

Methodological limitations arising from unmodeled variables (e.g., edaphic properties, anthropogenic land use changes) mandate integration with ground-truthing investigations. Periodic model recalibration (5–10-year intervals) using updated climate datasets enables dynamic recalibration of cultivation schemes, ensuring alignment with evolving bioclimatic realities. The predictive modeling framework integrating MaxEnt and ArcGIS establishes a scientific foundation for *Eucommia ulmoides* cultivation optimization in Southwest China. Future conservation strategies should prioritize monitoring precipitation variability and thermal regime alterations in core production zones, with coordinated implementation of adaptive silvicultural practices, genotype enhancement through selective breeding, and policy-driven regulatory frameworks to ensure sustainable resource utilization and industrial development.

## 5. Conclusions

This study employed the MaxEnt 3.4.4 species distribution model integrated with ArcGIS 10.8 to predict the potential habitats of *Eucommia ulmoides* Oliv. under SSP5-8.5 and SSP126 climate scenarios across four temporal intervals: 2021–2040, 2041–2060, 2061–2080, and 2081–2100. Analysis of geospatial distribution data and environmental variables revealed that current high-suitability habitats (probability >0.6) predominantly cluster in mid–low elevation valleys of the Daba Mountains and mesic zones of karst plateaus. Moderate-suitability areas extend across central Sichuan, eastern Guizhou, and marginally in eastern Yunnan. Optimal growth thresholds were quantified as precipitation of the driest month (12.6–96.5 mm), minimum temperature of the coldest month (−4–6 °C), and elevation range (13.9–756 m), confirming precipitation and thermal regimes as dominant bioclimatic determinants.

Under the high-emission scenario (SSP5-8.5), projections indicate a 23.7% contraction of high-suitability habitats by 2081–2100 compared to current conditions, coupled with a latitudinal migration trend toward higher latitudes. These findings collectively establish an ecological niche framework for guiding resource allocation, notably suggesting targeted afforestation in climate-resilient corridors (e.g., southern Chongqing karst basins) and genotype selection optimized for thermal tolerance gradients. The methodology provides theoretical guidance for sustainable resource utilization and precision cultivation protocols aligned with climate adaptation strategies in subtropical montane ecosystems.

## Figures and Tables

**Figure 1 biology-14-00451-f001:**
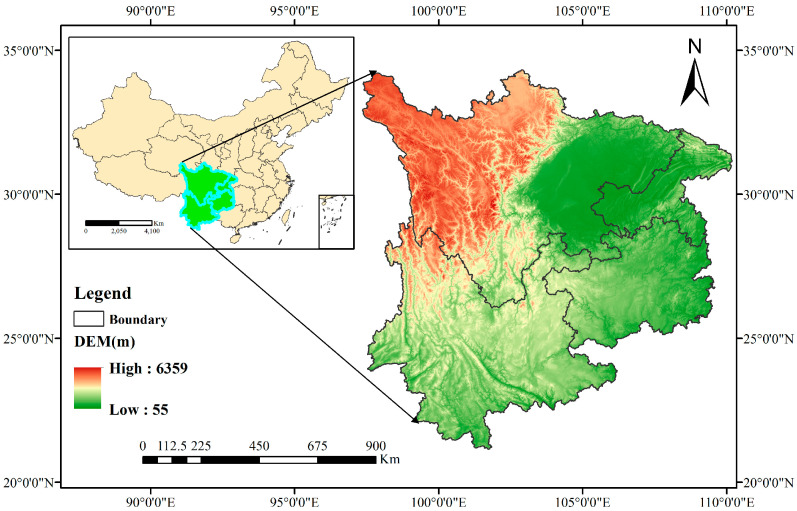
Overview of Southwest China.

**Figure 2 biology-14-00451-f002:**
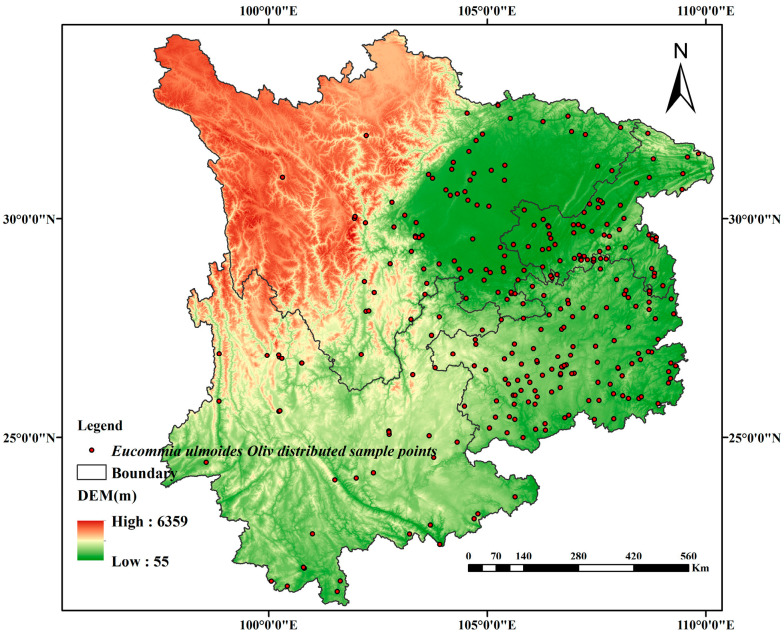
Site plot illustrating the distribution of *Eucommia ulmoides*.

**Figure 3 biology-14-00451-f003:**
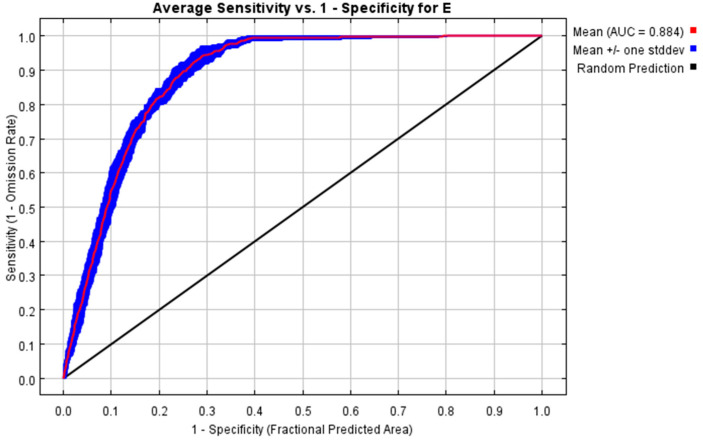
Receiver operating characteristic (ROC) curves for the same data.

**Figure 4 biology-14-00451-f004:**
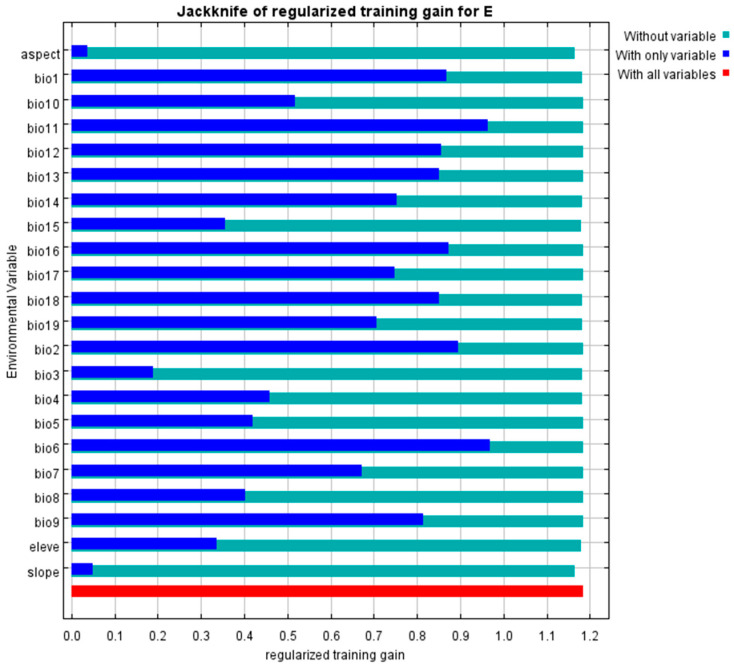
Test results of environmental variables.

**Figure 5 biology-14-00451-f005:**
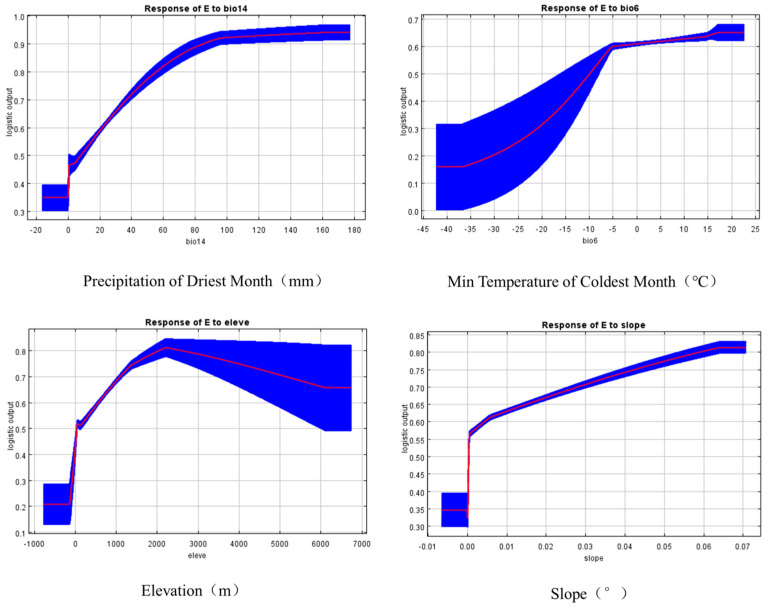
Response curves of the main environmental variable.

**Figure 6 biology-14-00451-f006:**
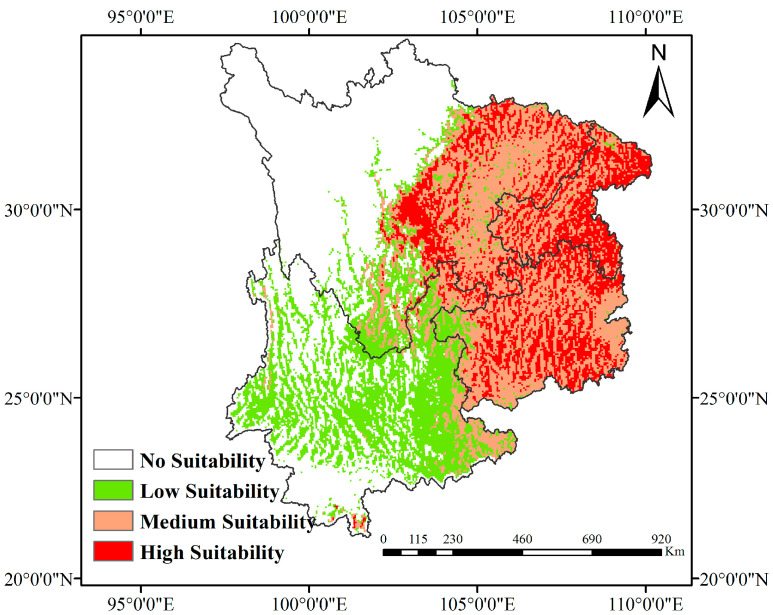
The potential suitable area for *Eucommia ulmoides* under current climate conditions.

**Figure 7 biology-14-00451-f007:**
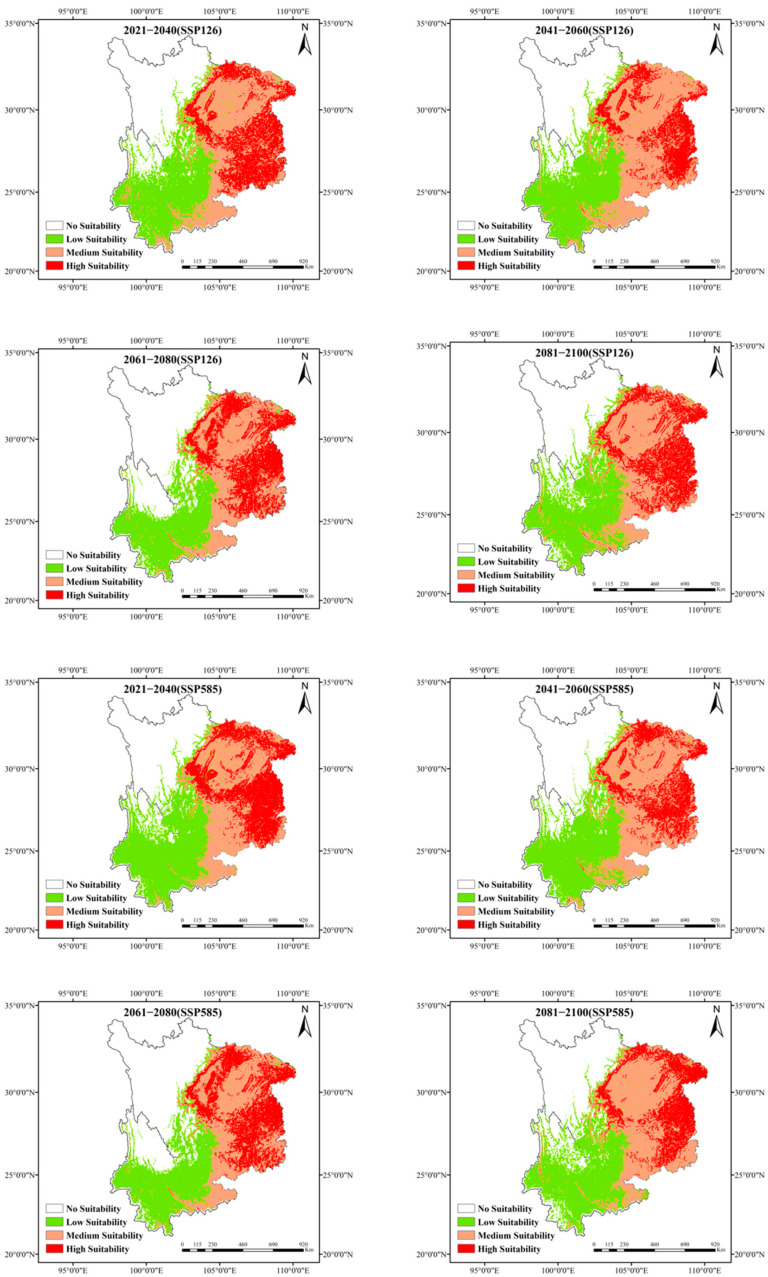
Projected suitable habitat range of *Eucommia ulmoides* under future climate conditions.

**Figure 8 biology-14-00451-f008:**
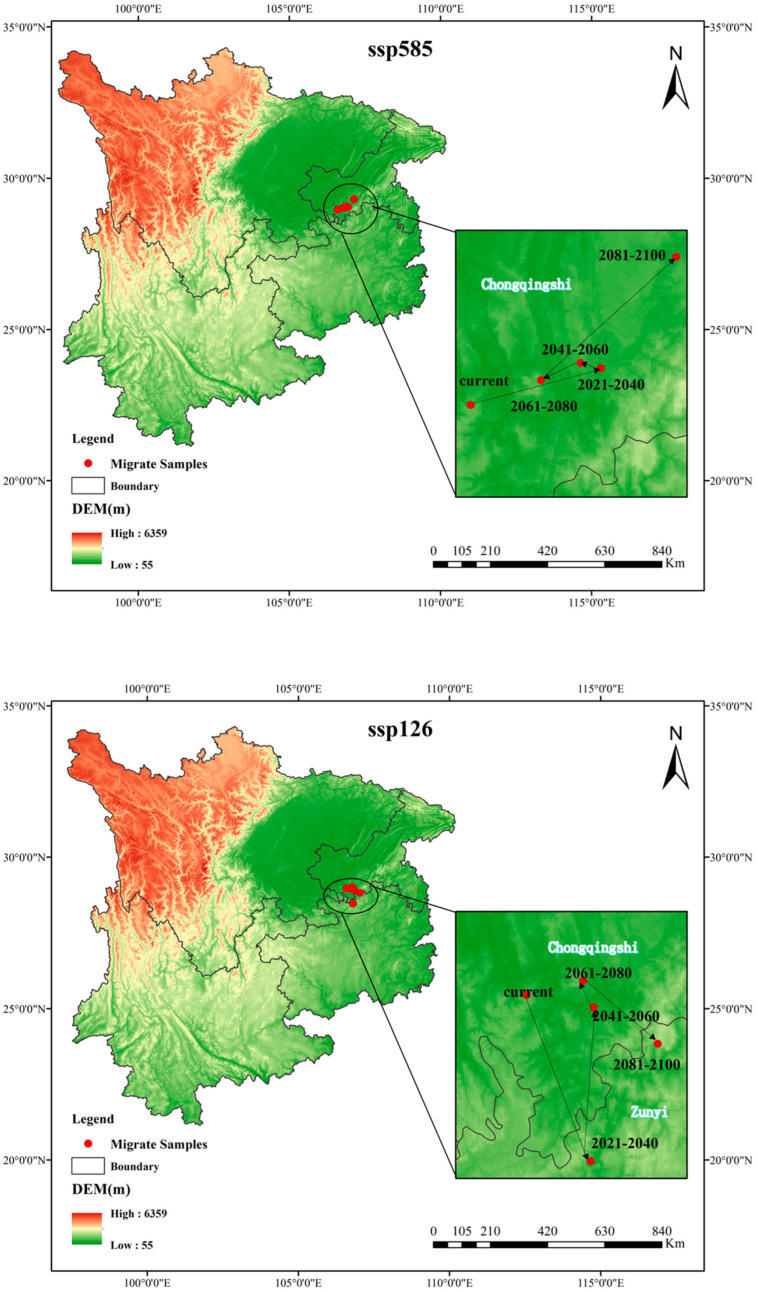
Geographic centroid displacement of high-suitability zones for *Eucommia ulmoides* under future climate conditions.

**Table 1 biology-14-00451-t001:** Twenty-two environmental variables.

Environment Variable	Description
bio1	Annual Mean Temperature (°C)
bio2	Mean Diurnal Range (Mean of monthly (max temp − min temp) (°C)
bio3	Isothermality (bio2/bio7) (×100) (bio2/bio7) (×100)
bio4	Temperature Seasonality (standard deviation ×100)
bio5	Max Temperature of Warmest Month (°C)
bio6	Min Temperature of Coldest Month (°C)
bio7	Temperature Annual Range (bio5–bio6)
bio8	Mean Temperature of Wettest Quarter (°C)
bio9	Mean Temperature of Driest Quarter (°C)
bio10	Mean Temperature of Warmest Quarter (°C)
bio11	Mean Temperature of Coldest Quarter (°C)
bio12	Annual Precipitation (mm)
bio13	Precipitation of Wettest Month (mm)
bio14	Precipitation of Driest Month (mm)
bio15	Precipitation Seasonality (Coefficient of Variation)
bio16	Precipitation of Wettest Quarter (mm)
bio17	Precipitation of Driest Quarter (mm)
bio18	Precipitation of Warmest Quarter (mm)
bio19	Precipitation of Coldest Quarter (mm)
elevation	Elevation (m)
aspect	Aspect
slope	Slope (°)

**Table 2 biology-14-00451-t002:** Environmental variables and their contribution rate screened by *Eucommia ulmoides* MaxEnt mode.

Environment Variable	Percent Contribution (%)	Permutation Importance (%)
bio14	27.2	5.90
bio6	22.1	11.0
bio13	14.90	0.40
bio12	10.40	7.10
bio11	4.20	20.0
slope	3.0	5.20
bio9	2.20	0.10
elevation	1.90	7.30

**Table 3 biology-14-00451-t003:** Thresholds of key environmental factors.

Environment Variable	Threshold
bio14	12.6 mm ≤ bio14 ≤ 96.5 mm
bio6	4 °C ≤ bio6 ≤ 6 °C
bio13	bio13 mm ≤ 184 mm; bio13 ≥ 345 mm
bio12	bio12 ≤ 216 mm; bio12 ≥ 970 mm
bio11	0.7 °C ≤ bio11 ≤ 9.8 °C; bio11 ≥18 °C
bio9	3 ≤ bio9 ≤ 11.7; bio9 ≥ 17
elevation	13.9 m ≤ elevation ≤ 756 m
slope	Slope ≥ 0.002°

**Table 4 biology-14-00451-t004:** Suitable habitat area of *Eucommia ulmoides* in Southwest China under future climate scenarios.

Suitable Area/km^2^					
Prediction Period		Low Suitable Area	Mid-Natural Area	High Fitness Area	Total Suitable Area
Current		9342	12,334	7255	28,931
SSP1-2.6	2021–2040	12,264	12,580	8411	33,255
	2041–2060	14,944	19,078	1916	35,938
	2061–2080	13,665	14,497	10,664	38,826
	2081–2100	14,458	16,103	10,082	40,643
SSP5-8.5	2021–2040	15,781	14,625	11,212	41,618
	2041–2060	15,552	17,605	8679	41,836
	2061–2080	13,662	14,480	10,633	38,775
	2081–2100	15,959	16,849	8594	41,402

## Data Availability

The original contributions presented in this study are included in the article. Further inquiries can be directed to the corresponding author.

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
