# Peer review of "Habitat Suitability Shifts of Eucommia ulmoides in Southwest China Under Climate Change Projections"

_biology, 2025, doi:10.3390/biology14040451_

Round 1
Reviewer 1 Report
Comments and Suggestions for Authors
Thank you for the opportunity to review your manuscript, "Habitat Suitability Shifts of Eucommia ulmoides in Southwest China under Climate Change Projections." Your study presents an important and timely investigation into the potential distribution shifts of Eucommia ulmoides under future climate scenarios, using MaxEnt modeling and geospatial analysis. The integration of species distribution modeling (SDM) with climate projections (SSP1-2.6 and SSP5-8.5) provides valuable insights into how environmental changes may impact this ecologically and economically significant species. The approach is robust, combining multiple temporal projections and spatial analysis through ArcGIS, which strengthens the reliability of the findings. Additionally, your identification of key environmental drivers, such as precipitation seasonality, thermal extremes, and elevation, adds further depth to the discussion.
While the manuscript has strong merits, there are areas that require improvement to enhance clarity, coherence, and methodological transparency. One primary concern is the complexity of the writing style, which at times is overly technical and could be streamlined to improve readability. Some sections, particularly in the introduction, could be more concise by avoiding redundant explanations of climate change impacts and instead focusing more directly on the justification for studying E. ulmoides. A clearer articulation of the research gap your study addresses would strengthen the manuscript’s impact.
Additionally, figures and tables could also be better integrated into the text, ensuring that they are consistently referenced and positioned logically for improved flow. Furthermore, the manuscript frequently introduces acronyms (e.g., AUC, SSP, ROC, SDM) without always defining them on first use, which could make it more challenging for a broader audience to follow.
From a methodological perspective, the manuscript would benefit from a more detailed discussion of model calibration. While you mention the use of default parameters in MaxEnt, it would be valuable to clarify whether any specific tuning was performed to optimize model performance. Moreover, while the study provides an evaluation metric (AUC), the discussion could be expanded to acknowledge the limitations of the MaxEnt model, such as its dependence on occurrence record quality, potential biases from spatial autocorrelation, and the resolution of climate data. Addressing these uncertainties would improve the robustness of your conclusions.
Lastly, the discussion section could include a more explicit assessment of potential conservation strategies derived from your findings. While you mention the implications for silviculture and agroforestry, further elaboration on how this information could inform conservation planning, policy, or adaptive management would strengthen the study’s real-world applicability.
Comments on the Quality of English Languageneed to be improved in some parts
Author Response
Comment 1: One primary concern is the complexity of the writing style, which at times is overly technical and could be streamlined to improve readability. Some sections, particularly in the introduction, could be more concise by avoiding redundant explanations of climate change impacts and instead focusing more directly on the justification for studying E. ulmoides. A clearer articulation of the research gap your study addresses would strengthen the manuscript's impact.
Response: Thank you for highlighting this issue. We have streamlined the discussion of climate change impacts in the Introduction section, enhancing its focus (revised manuscript, page 2). A new section on the ecological adaptability of Eucommia ulmoides has been added, supported by [specific data/references], to strengthen the empirical foundation of the analysis.
Comment 2: Figures and tables could also be better integrated into the text, ensuring that they arconsistently referenced and positioned logically for improved flow.
Response: Thank you for raising this concern. We have added thresholds of key environmental factors and future suitable habitat area charts to the revised manuscript (pages 7 and 8), with these modifications highlighted for clarity.
Comment 3: The manuscriptfrequently introduces acronyms (e.g., AUC, SSP, ROC, SDM) without always defining them onfirst use, which could make it more challenging for a broader audience to follow.
Response: Thank you for raising this concern. We have clarified key technical terms in the Introduction and Section 2.4 to improve conceptual clarity (page 5).
Comment 4: While you mention the use of default parameters in MaxEnt, itwould be valuable to clarify whether any specific tuning was performed to optimize modelperformance.
Response: Thank you for highlighting this issue. No further refinement of the model was implemented in the current study.
Comment 5: Moreover, while the study provides an evaluation metric (AUC), the discussioncould be expanded to acknowledge the limitations of the MaxEnt model, such as its dependenceon occurrence record quality, potential biases from spatial autocorrelation, and the resolution ofclimate data. Addressing these uncertainties would improve the robustness of your conclusions.
Response: Thank you for raising this concern. We have included an analysis of the MaxEnt model’s limitations on page 12 of the revised manuscript
Comment 6: The discussion section could include a more explicit assessment of potential conservationstrategies derived from your findings. While you mention the implications for silviculture andagroforestry, further elaboration on how this information could inform conservation planning.policy, or adaptive management would strengthen the study's real-world applicability.
Response: Thank you for highlighting this issue. Specific cultivation strategies for Eucommia ulmoides have been proposed based on the research findings (page 12).
Reviewer 2 Report
Comments and Suggestions for Authors
This paper investigates the habitat suitability shifts of Eucommia ulmoides in southwest China under climate change projections.this analysis establishes an evidence-based foundation for optimizing silvicultural practices and adaptive resource governance under evolving climatic regimes. However, there still are some suggestions to improve the manuscript.
Q1: Keywords. This paper focuses solely on one medicinal plant; therefore, this keyword using the species name 'Eucommia ulmoides' may be more appropriate.
Q2: In “Graphical Abstract”, the yellow text does not seem clear, it is recommended to change the font or color
Q 3: “2.3. Collection and Evaluation of Environmental Data” Most of the environmental variables are precipitation and temperature, and they are divided into several different variables, which I think will affect the subsequent results.The authors should treat precipitation and temperature as the same as latitude and slope, in order to truly identify which factor is the key to species distribution. Of course, different climate scenarios can be used to predict species distribution in the context of climate change
Q 4: L140:”four habitat suitability index (P) categories“,Whether there is any basis or reference for this classification needs to be explained.
Q 5: L172、186..., Many paragraphs of the article begin with "Figure 3","Figure 4" etc., which I don't think is appropriate
Q 6: L246:reference.
Q 7: L311: Why are important factors such as "edaphic proper-ties, anthropogenic land-use changes" not included in climate prediction models
Author Response
Comment 1: Keywords. This paper focuses solely on one medicinal plant; therefore, this keyword usingthe species name 'Eucommia ulmoides' may be more appropriate.
Response: Thank you for raising this concern. We have clarified key technical terms in the Introduction and Section 2.4 to improve conceptual clarity.
Comment 2: In "Graphical Abstract", the yellow text does not seem clear, it is recommended to change thefont or color.
Response: Thank you for raising this concern. The yellow font in the graphical abstract has been changed to red to improve visual clarity.
Comment 3: “2.3. Collection and Evaluation of Environmental Data” Most of the environmental variables are precipitation and temperature, and they are divided into several different variables, which I think will affect the subsequent results.The authors should treat precipitation and temperature as the same as latitude and slope, in order to truly identify which factor is the key to species distribution. Of course, different climate scenarios can be used to predict species distribution in the context of climate change.
Response: Thank you for raising this concern. In climate simulations, temperature (driving the energy cycle) and precipitation (driving the hydrological cycle) are fundamental variables for Earth system analysis. These variables directly influence ecological, agricultural, and hydrological studies. Most climate models prioritize these widely applicable variables in their default outputs to address diverse research needs, such as climate change assessments and extreme event analyses.
Comment 4: L140:”four habitat suitability index (P) categories“,Whether there is any basis or reference for this classification needs to be explained..
Response: Thank you for raising this concern. We have supplemented the details under Section 2.5 (page 5) of the revised manuscript. Zone classification was based on two criteria: Maximum Training Sensitivity plus Specificity (MTSS) and the Balance between Training Omission, Predicted Area, and Threshold Value (BTPT).
Comment 5: L172、186..., Many paragraphs of the article begin with "Figure 3","Figure 4" etc., which I don't think is appropriate.
Response: Thank you for raising this concern. We have revised the manuscript accordingly, with modifications highlighted in yellow across pages 6–10 for transparent tracking.
Comment 6: L246:reference.
Response: Thank you for highlighting this issue. We have revised the References section accordingly.
Comment 7: L311: Why are important factors such as "edaphic proper-ties, anthropogenic land-use changes" not included in climate prediction models.
Response: Thank you for raising this concern. Anthropogenic activities and soil impacts within the climate system remain subject to uncertainties, primarily due to limitations in observational data, modeling frameworks, and process understanding. Furthermore, critical research gaps persist in quantifying anthropogenic contributions to key climatic indicators (e.g., atmospheric, oceanic, cryospheric, biospheric, and climate variability metrics).
Round 2
Reviewer 1 Report
Comments and Suggestions for Authors
Thank you for the thoughtful and detailed revisions made to the manuscript. I am satisfied with the authors’ responses and the improvements in clarity, structure, and integration of figures and discussion. The incorporation of ecological context and conservation strategies greatly enhances the relevance of the study.
However, I would kindly suggest that the authors also address Comment 4 more thoroughly in the manuscript itself. While the response clarifies that default MaxEnt parameters were used, the manuscript would benefit from a brief discussion on the implications and potential limitations of this choice. For instance, commenting on how the lack of parameter tuning might affect model performance (e.g., overfitting or underfitting), and whether this was considered a limitation or an acceptable trade-off in the context of this study, would strengthen the methodological transparency.
Author Response
Comment : I would kindly suggest that the authors also address Comment 4 more thoroughly in the manuscript itself. While the response clarifies that default MaxEnt parameters were used, the manuscript would benefit from a brief discussion on the implications and potential limitations of this choice. For instance, commenting on how the lack of parameter tuning might affect model performance (e.g., overfitting or underfitting), and whether this was considered a limitation or an acceptable trade-off in the context of this study, would strengthen the methodological transparency..
Response: We appreciate your identification of this issue. The model specifications, including detailed data parameters, are comprehensively addressed in Section 2.4 (page 5). Methodological constraints are systematically analyzed within the Discussion section (page 12). All corresponding revisions have been marked with yellow highlights to streamline the editorial evaluation process.
We appreciate your thorough review. We sincerely regret the oversight and have implemented all advised revisions. The updated manuscript is respectfully resubmitted for your further evaluation.
Qi Liu
Guizhou University of Traditional Chinese Medicine, Guiyang 550025, China
6 April , 2025